# A Plausible Framework Reveals Potential Similarities in the Regulation of Immunity against Some Cancers and Some Infectious Agents: Implications for Prevention and Treatment

**DOI:** 10.3390/cancers16071431

**Published:** 2024-04-07

**Authors:** Peter A. Bretscher

**Affiliations:** Department of Biochemistry, Microbiology & Immunology, College of Medicine, University of Saskatchewan, Saskatoon, SK S7N 5A2, Canada; peter.bretscher@usask.ca

**Keywords:** cancer immunotherapy, vaccination against cancer, increasing cancer immunogenicity, modulating the class of the anti-cancer response, concomitant immunity, monitoring of the anti-cancer response

## Abstract

**Simple Summary:**

The immune system can fight foreign invaders by different means, expressed by different classes of immunity. There are two main classes: antibodies and cell-mediated immunity. Studies directed at how distinct classes of immunity are differentially generated and regulated have been ongoing for the better part of a century. Various quantitative variables of immunization, such as antigen doses, are critical, and were first identified more than fifty years ago. It was shown that lower and higher doses, respectively, lead to cell-mediated and antibody responses. The importance of these variables has stood the test of time. We argue here for a framework, the Threshold Hypothesis, that is consistent with these variables and accounts for their central role in determining the class of immunity generated. This framework leads to an understanding of many observations in the fields of tumor immunology and immunity against pathogens uniquely contained by cell-mediated attack. This understanding relies on the generalization that cancers are uniquely susceptible to cell-mediated attack. This confidence in the threshold mechanism led me to propose non-invasive and efficacious strategies to prevent and treat cancer and infectious diseases caused by pathogens uniquely susceptible to cell-mediated immunity.

**Abstract:**

Different frameworks, which are currently employed to understand how immune responses are regulated, can account for different observations reported in the classical literature. I have argued that the predominant frameworks, employed over the last two/three decades to analyze the circumstances that determine whether an immune response is generated or this potential is ablated, and that determine the class of immunity an antigen induces, are inconsistent with diverse classical observations. These observations are “paradoxical” within the context of these frameworks and, consequently, tend to be ignored by most contemporary researchers. One such observation is that low and high doses of diverse types of antigen result, respectively, in cell-mediated and IgG antibody responses. I suggest these paradoxes render these frameworks implausible. An alternative framework, The Threshold Hypothesis, accounts for the paradoxical observations. Some frameworks are judged more plausible when found to be valuable in understanding findings in fields beyond their original compass. I explore here how the Threshold Hypothesis, initially based on studies with chemically well-defined and “simple antigens”, most often a purified protein, can nevertheless shed light on diverse classical and more recent observations in the fields of immunity against cancer and against infectious agents, thus revealing common, immune mechanisms. Most cancers and some pathogens are best contained by cell-mediated immunity. The success of the Threshold Hypothesis has encouraged me to employ it as a basis for proposing strategies to prevent and to treat cancer and those infectious diseases caused by pathogens best contained by a cell-mediated attack.

## 1. Prospect: The Impetus for Writing This Paper

All immunologists are aware of the progress in their field and of its potential impact on global health problems. This potential significance has been recognized by substantial increases in public investment in immunological research [1]. It has been convincingly argued that the increased investment in diverse scientific fields, since WWII, has not resulted in unalloyed progress [2,3,4]. It seems to be recognized that the increased rate of production of information has not allowed researchers the peace to sufficiently consider the significance of the new information. The situation has been described by saying the canon of the field has become ossified [2]. Many consider this lack of research resilience to be an inevitable consequence of the information overload [2,3,4]. 

I argue that a consequence of the ossification of contemporary frameworks is an accumulation of paradoxes within their context [5]. The endeavor to seek out and resolve paradoxes within the context of the contemporary frameworks can result in their evolution and their broader pertinence to allied fields. This increasing breadth of the pertinence of a framework is what makes, at least some, believe in its value and contemporary validity. The unabated ossification of the canon, in contrast, results in research silos developing in neighboring and related fields [5].

I have argued elsewhere that the predominant frameworks employed to understand how the class of immunity against an antigen is determined are implausible, because, paradoxical with many classical observations, primarily made in basic studies on how immune responses are regulated with experimental antigens [5]. In addition, since these frameworks have been insufficiently challenged by the immunological community, they have remained remarkably static over the last two/three decades. It is of significant importance that some classical observations, those that are paradoxical in the context of the dominant, contemporary frameworks, are often forgotten and/or ignored [5]. I have again and recently argued that those observations, which are paradoxical in the context of the dominant frameworks, are accounted for by an alternative framework [5] that, at its core, is quantitative. This framework thus brings quantitative considerations to the fore. My purpose in this article is to explore how the alternative framework I favor also provides a common basis for understanding some classical and more recent observations in infectious diseases and tumor immunology that have also been, in my opinion, relatively ignored. To provide context, I shall first provide a synopsis of why I judge the dominant frameworks to be implausible in contrast to the alternative framework. I provide this synopsis in point form for ease of reference.

## 2. Non-Contentious Issues That Provide the Context for Discussing Contemporary Issues That Should Be Challenged 

Classical studies gave rise to the view that there are central and peripheral mechanisms of tolerance against self-antigens [6]. Self-antigens, when sufficiently present in primary lymphoid organs where lymphocytes are generated, cause the ablation of their corresponding lymphocytes [7] by a mechanism first proposed by Lederberg [8]. This mechanism accounts for central tolerance. Some self-antigens, of which insulin is a prototype, are insufficiently present in the primary lymphoid organs to ablate all the lymphocytes specific for these antigens [9]. Such self-antigens are called peripheral self-antigens. It is recognized that there must be a mechanism by which peripheral self-antigens can inactivate their corresponding lymphocytes, which emigrate from primary lymphoid organs, in order to ensure self-tolerance [6]. This inactivation process results in peripheral tolerance.The two-signal model of lymphocyte activation, which proposes how antigen can activate and inactivate mature, naïve lymphocytes, was proposed in 1970 [10]. This theory posited that the activation of a target lymphocyte required its antigen-mediated interaction with a helper lymphocyte. The activation of all lymphocytes thus requires lymphocyte cooperation. Antigen inactivates single lymphocytes. The theory was proposed because it accounted for diverse observations, some paradoxical at the time, and because it accounted for how peripheral tolerance can be achieved. This latter feature is germane to the current considerations.The theory explained peripheral tolerance in the context of an idea first proposed by Burnet and Fenner [11] and later extended by Lederberg [9]. The extended idea was that tolerance to self-antigens requires their early presence in development, or in the *history* of the individual, before lymphocytes are generated, and their continuous presence thereafter. This idea is referred to as The Historical Postulate. Thus, lymphocytes specific for peripheral self-antigens are inactivated as they are generated, one or a few at a time, by virtue of the early and continuous presence of the peripheral self-antigen. Lymphocytes specific for a foreign antigen, F, accumulate in its absence; once F impinges upon the immune system, it can mediate the interactions between the accumulated lymphocytes required to generate an immune response [12].It is convenient to refer to the signal generated from the interaction of antigen with the antigen-specific receptors of the lymphocyte as signal 1, and the delivery of the signal to the target lymphocyte, following the recognition of antigen by the helper lymphocyte, as signal 2. The generation of signal 1 alone is envisaged to result, in time, in the inactivation of the lymphocyte. The activation of the target lymphocyte requires the generation of both signal 1 and of signal 2 [10]. This theory is often referred to as the two-signal model of lymphocyte activation. However, there are other two-signal models of lymphocyte activation, as I make clear below.Much evidence supports this two-signal model. Thus, it is generally accepted that the activation of most B cells [13,14,15,16] and CD8 T cells [17], to, respectively generate antibody-producing cells and cytotoxic T lymphocytes (CTL), requires activated CD4 T helper cells. In the absence of this help, antigen inactivates the B [18,19] and the CD8 T cells [20]. Such observations, made over the years, have led virtually all immunologists to recognize the critical importance of activating and inactivating CD4 T helper cells. The central importance of helper T cells is acknowledged by referring to them as “the conductor of the immunological orchestra”. This topic of how antigen activates and inactivates CD4 T cells is the gate to controversy. It became evident by the late 1970s that the antigen-specific receptors of T cells recognize peptides, derived by processing the nominal antigen, bound to class I or II MHC molecules [21,22,23]. Antigen-presenting cells (APC) had the means of taking an antigen up, processing it, and presenting it [24]. Various discoveries led to the realization that the APC, in order to activate the CD4 T cell, had to deliver a co-stimulatory signal to the CD4 T cell. This required the APC to bear costimulatory molecules and the CD4 T cell to bear counter receptors. It was envisaged that the CD4 T cell was inactivated in the absence of costimulatory signals [25,26,27]. An interaction between the B7 molecules on the APC, and their counter receptors, CD28, expressed by CD4 T cells, represents the prototypical costimulatory (CoS) signal [28].

## 3. The PAMP/DAMP-Centric View on How Immune Responses Are Regulated

Janeway suggested in 1989 that APC had to be activated to express sufficient CoS molecules to allow antigen to activate CD4 T cells [29]. He proposed that such activation occurred when pathogen-associated molecular patterns (PAMP) interact with pathogen recognition receptors (PRR) [30,31]. A major consideration underlying this proposal was the claim that foreign, vertebrate antigens were not by themselves immunogenic unless given with adjuvants containing material that expressed one or more PAMPs. In contrast, entities containing materials that expressed PAMPs were highly immunogenic [29]. Matzinger, five years later, pointed out that skin grafts between mice belonging to different strains are rejected, an observation she felt to be incompatible with Janeway’s model, as mouse grafts themselves would not express novel PAMPs [32]. Grafting involves the graft-recipient being under considerable stress, most likely resulting in the expression of stress responses. Matzinger proposed that instead of a PAMP signal, the full expression of CoS molecules by APC requires the expression and recognition of a danger-associated molecular pattern, i.e., a DAMP or danger signal [32,33,34]. It is now generally believed by most immunologists that CD4 T cells can be inactivated by antigen in the absence of a DAMP/PAMP signal. Matzinger’s Danger and Janeway’s PAMP Theory are the dominant, contemporary theories. Most researchers believe that these proposals are also relevant to how the class of immunity is determined [31,33,34]. Local factors, dependent on the route of immunization, are also known to be important [35]. It has therefore been proposed and generally accepted that the particular nature of the DAMP/PAMP signals is critical in determining the class of immunity generated [31,33,34,36,37]. These are the dominant, contemporary frameworks. I collectively refer to these ideas as the PAMP/DAMP-centric view. 

## 4. The Variables of Immunization Known to Affect the Th1/Th2 Phenotype of the Ensuing Immune Response and the Physiological Significance of the Class of Immunity Generated 

If we had a correct description of how antigens interact differently with the cells of the immune system, activating naïve CD4 T cells to generate Th1 and Th2 cells, we would be able to explain why different circumstances of immunization give rise to Th1 and Th2 responses [38]. Thus, we need to know what these variables of immunization are if we are to judge whether ideas on the nature of the decision criterion, controlling the Th1/Th2 phenotype of a response, are plausible. There are several variables of immunization that first appeared to be significant in studies carried out over 50 years ago. Time has only reinforced the importance of these variables. I have “transcribed” these older observations into contemporary terms. 

(a)Pearson and Raffel argued, on the basis of observation, that minimally foreign antigens, due either to their small size or being larger but only varying slightly from a corresponding self-antigen, are only able to generate Th1, delayed-type hypersensitivity (DTH) responses [39].(b)Salvin showed that immune responses evolve with time from an exclusive Th1 mode to one containing a significant Th2 component with IgG antibody production [40], see Figure 1. The response sometimes evolves into a predominant Th2 mode.(c)Salvin also examined the effect of antigen dose [40]. Low doses can generate an exclusive Th1, DTH response; see Figure 1. Medium doses result in the more rapid generation of a Th1 response and, with time, Th2 cells appear and copious IgG antibody is produced. Even higher doses result in an even more rapid or transient Th1 response, and in a predominant Th2 response.(d)Immunization of an animal in a manner that results in an IgG antibody humoral response renders the animal unable to generate a DTH Th1 response to a challenge that generates such a response in a naïve animal [42]. The animal’s response to this antigen appears to have been locked into a humoral, IgG antibody mode. This phenomenon is referred to as “humoral immune deviation” [5]. Parish showed that repetitive immunization, over several weeks, of animals with a low dose of antigen, sub-immunogenic for an antibody response, resulted in a state of DTH, which is now recognized as being associated with the generation of Th1 cells. These exposed animals were partially unresponsive/unresponsive for the production of antibody [43,44]. This phenomenon is referred to as “cell-mediated immune deviation”.(e)It was first recognized, in studies on immunity of leprosy patients, that the class of immunity generated against the pathogen largely determines the course of disease. In this case, the generation of a stable and predominant Th1 cell-mediated response is associated with minimal disease, and mixed and predominant antibody responses with distinct but severe pathologies [45]. In more recent decades, the mouse model of the human disease of cutaneous leishmaniasis has become the major experimental model for a disease caused by a pathogen preferentially contained by cell-mediated attack [46]. The pertinent intracellular, protozoan pathogen, *Leishmania major*, inhabits macrophages. We will discuss below several studies conducted using this experimental system.

## 5. The Threshold Hypothesis Explains How the Th1/Th2 Phenotype of the Response Is Determined

I proposed this hypothesis in the early 1970s, largely because it accounts for the variables of immunization then known to affect the Th1/Th2 phenotype of the ensuing response [38]. I have recently summarized the evidence that supports the hypothesis [5]. Here, I give the briefest outline of it. This provides a context so that we can explore its utility in understanding immunity against cancer and against some infectious pathogens.

Ideas as to how the Th1/Th2 phenotype of a response is determined are inevitably based upon what one envisages are the requirements to activate CD4 T cells. The PAMP/DAMP-centric view implies that what controls whether an antigen inactivates or activates a CD4 T cell does not depend on whether it is a peripheral self-antigen or a foreign antigen, but rather whether the impact of the antigen upon the immune system is associated with a DAMP or a PAMP signal, as outlined above. This was a radical suggestion when first made and it still is, as far as I am concerned. There are many observations and considerations that make me doubt it [5]. One is that there are many foreign, vertebrate, and so PAMP-free antigens that are highly immunogenic when an antigen is delivered with a sharp needle, in the absence of PAMP-containing adjuvants. One such antigen, used in many classical studies, is sheep red blood cells (SRBC) administered to mice [47]. I favor a more detailed form [48] of the two-signal model described in 1970 [10] for the activation and inactivation of CD4 T cells. The activation of a target CD4 T cell is envisaged to require its antigen-mediated cooperation with other helper CD4 T cells, with an antigen-specific B cell acting as the intermediary APC. In the absence of CD4 T cell cooperation, the CD4 T cell is inactivated by the antigen. This model incorporates the idea that the immune system does discriminate peripheral self-antigens and foreign antigens at the level of CD4 T cells, in a manner consistent with The Historical Postulate. This makes it the conservative alternative among contemporary proposals. As noted, I have recently reviewed the substantial evidence in support of it [12].

The Threshold Hypothesis assumes that the activation of naïve CD4 T cells requires CD4 T cell cooperation. It posits that when antigen-mediated CD4 T cell cooperation is weak, Th1 cells are generated. When it is strong, Th2 cells are generated. This hypothesis accounts for all the quantitative variables of immunization known to affect the Th1/Th2 phenotype of the ensuing response and listed above [38]. It accounts for Pearson and Raffel’s observation, that minimally foreign antigens can only generate Th1 cells. The number of CD4 T cells specific for such antigens is few compared to the number specific for more foreign antigens. Thus, even in the presence of optimal levels of antigen to mediate CD4 T cell cooperation, such cooperation will be weak and only Th1 cells will be generated. I also proposed that this type of chronic stimulation results in the response becoming locked into a cell-mediated mode, due the accumulation of T cells that inhibit the ability of the CD4 T cells to generate Th2 cells. This proposal was made to explain how cell-mediated immune deviation could be established. There are more CD4 T cells specific for more foreign antigens in a naïve animal than for a minimally foreign antigen. When antigen activates CD4 T cells, it causes them to divide. If the antigen level is sufficiently sustained, CD4 T cell cooperation will become stronger with time. Thus, a very low dose of a more foreign antigen will result in weak cooperation and the generation of Th1 cells. Such chronic stimulation will result in cell-mediated immune deviation, as just described. Immunization with substantially higher doses results in faster CD4 T cell activation and so the faster generation of Th1 cells. As the CD4 T cells divide on antigen impact, the strength of CD4 T cell interactions may increase sufficiently so that Th2 cells are generated. If the antigen dose is further increased, and so even more optimal for mediating CD4 T cell cooperation, the tempo of responses further increases. The Th1 phase may even become transient [38]. The proposed threshold mechanism thus accounts for Salvin’s observations; see Figure 1. The Threshold Hypothesis makes one particularly striking prediction. Consider a situation where a response with a substantial Th2 component is generated. The threshold mechanism predicts that the partial depletion of CD4 T cells, at the time of immunization, will result in a modulation of the response towards a Th1 mode. This prediction has been successfully tested in different and diverse experimental systems, as described elsewhere [5,49]. One particular case is well known and particularly interesting. BALB/c mice are regarded as susceptible to *L. major* on the basis that they rapidly develop a Th2 response to infection with the standard challenge of a million parasites, and so suffer uncontrolled parasitemia and progressive disease. Partial depletion of CD4 T cells around the time of infection results in a stable Th1 response, control of parasitemia and so in resistance [50,51]. 

## 6. The Plausibility/Implausibility of the PAMP/DAMP View 

I have described above how the PAMP/DAMP-centric view bears on both the activation of CD4 T cells and their differentiation to generate Th cells belonging to different Th subsets. I will not discuss the pertinence of this view to the activation of CD4 T cells here, as this has been both recently discussed [12] and is not as central to the topic at hand as the differentiation fate of CD4 T cells into different Th subsets. I list below, in point form, the major reasons for considering the PAMP/DAM-centric view inadequate for understanding the basis of how the Th1/Th2 phenotype of the response is determined. 

It is known that the dose of antigen is important for determining the Th1/Th2 phenotype of the response in the manner outlined above, with lower doses favoring Th1 responses. This is true of vertebrate, PAMP-free proteins, administered with adjuvant containing PAMPs [40]; of more complex, foreign, PAMP-free vertebrate antigens, administered without adjuvant, such as SRBC in mice [47]; of protozoa administered to mice [52,53]; and of mycobacteria administered to mice [5] and to cattle [54]. The protozoa and mycobacteria clearly bear very different PAMPs. It seems likely, given the generality of this dependence of the class of immunity on antigen dose, generated in response to diverse types of antigen, that a PAMP-independent mechanism is likely critical, as envisaged by the threshold mechanism [5,38].Most immune responses evolve from an exclusive Th1 mode to one with a significant Th2 component. This is true of foreign, vertebrate, PAMP-free protein antigens, administered in PAMP-containing adjuvants [40]; of foreign, vertebrate, PAMP-free, and chemically complex antigens, administered without adjuvant, such as SRBC in mice [46]; of protozoa administered to mice [52,53]; and of mycobacteria administered to mice [5] and cattle [54]. Again, the generality of this pattern in responses to such diverse antigens leads to the idea that it can only be accounted for by a PAMP/DAMP-independent mechanism. Such a mechanism is provided by The Threshold Hypothesis.We have briefly outlined “the CD4 T cell depletion experiment” above, showing that partial depletion modulates the response to decrease the Th2 and increase the Th1 component. Most of these observations were designed to test a critical prediction of the threshold mechanism [5,49]. However, such experiments also provide evidence against the PAMP/DAMP-centric view. No proponents of this view have explained how the number of CD4 T cells can affect the nature of the PAMP or DAMP signal. Thus, these observations present problems for the PAMP/DAMP-centric view. Again, these observations support the threshold mechanism.

Most of these CD4 T cell-depletion experiments that we carried out employed foreign, vertebrate, PAMP-free antigens in diverse experimental systems [5,49]. The convincing nature of this collective body of evidence is in part due to the diversity of the different circumstances under which the prediction holds. We discuss other studies below in greater detail; these involve CD4 T cell depletion experiments in mouse systems of infectious disease and of tumor immunology.

## 7. Consequences of the Popularity of the PAMP/DAMP-Centric View

I must admit I have been somewhat puzzled by the fact that quite a number of older studies, as well as some more recent ones, that I feel are highly significant, have little impact on current views. I have concluded, after considerable reflection, that the frameworks one favors govern not only one’s interpretation of observations, but also affect one’s judgement of what observations are significant. The PAMP/DAMP-centric views are, in my view, rather open-ended; they do not make precise predictions. There are no quantitative features of these ideas. Many of the neglected observations have quantitative features, as is hopefully evident from the considerations outlined above. The Threshold Hypothesis is, at its core, quantitative. The above pages have been written in the hope of convincing a reader, who has been inclined to adhere to the predominant, contemporary views, to be open to reconsidering older studies that may provide insights.

## 8. The View of Tumor Immunology Prevalent in the Late 1960s and Early 1970s

I remember the predominant views holding sway in tumor immunology in this past era, based, of course, on rather simple animal experiments. However, it is important to acknowledge upfront the potential limitations of the pertinence to human cancer of such animal studies. 

Most studies in animals involve examining immune responses to tumor cell lines. These lines had been established from either spontaneous tumors, tumors caused by infection of cells with oncogenic viruses, or those arising from administering carcinogens to animals. The resulting tumors were usually passaged in vitro, to establish cell lines that are relatively stable and can be employed to generate relatively repeatable observations. Such situations are a far cry from cancers spontaneously arising in humans; these likely go through several stages of development to establish a progressive cancer [55]. I personally consider that objections to drawing conclusions derived from animal studies when considering how to control and treat human cancer have considerable force. My view is that we should try to understand what transpires in such animal studies and exert due circumspection in drawing inferences pertinent to the prevention or treatment of human cancer. I try to follow this path. 

Three generalizations were made in the 1960s and 1970s era that are of considerable interest. Firstly, means were found to render animals resistant to an otherwise lethal tumor challenge. We shall return shortly to the nature of some of these means. Such studies led to the idea that tumor resistance is associated with a predominant cell-mediated response, and susceptibility is often associated with the production of antibody and the associated downregulation of the cell-mediated response. I cannot do better to summarize the views then prevalent than by quoting from an address given by George Klein [56], one of the founders of tumor immunology. He said in his 1968 address, in the context of vaccination against tumors: “It will be most important to establish what variables of … immunization … dosage, route of administration and timing are critical to achieve a stimulation of host cell-mediated immunity with minimum risk of antibody-mediated enhancement”. This statement has the optimism of a pioneer. We shall later consider whether this view was hopelessly naïve. 

A second generalization concerns a procedure by which resistance to a normally lethal challenge of the tumor could be realized. A normally lethal challenge of the tumor was inoculated into the animal at a surface site where the growth of the tumor could be readily monitored. This tumor was operated out, when about a millimeter or two in size, about ten days after implantation [57]. Such “tumor-excised” mice were found to resist a normally lethal change, given some considerable time—a month or two—after tumor excision. It is natural to wonder what processes underly the acquirement of this resistance to a normally lethal challenge. However, I do not think this question has been adequately discussed and a consensus has yet to be agreed upon within the immunological community. I shall return to this question.

The third generalization was indeed intriguing. Animals were again given a lethal challenge of a tumor at a primary, surface site. A second and identical challenge of the same tumor, normally lethal, was given at a distal, surface site, about ten days later. The tumor at the first site grew progressively, but in most cases the second challenge did not. This effect was tumor specific. The growth of a different type of tumor at the second site was not impeded. It was concluded that the implantation of a lethal tumor challenge resulted, in about ten days after implantation, in protective immunity, resulting in resistance against the second challenge. Such immunity was called concomitant immunity [58]. The question arose as to why the primary tumor grew progressively in face of this concomitant immunity. This question inspired Robert North.

## 9. Robert North’s Studies

North was convinced that an understanding of why a lethal dose of tumor cells was not eliminated by concomitant immunity was critical; he expressed the hope that such understanding would allow the development of strategies to harness immunity to fight cancer. His enthusiasm was evident.

North brought many observations together in the 1980s in developing his picture of what happens immunologically when an animal is given a lethal tumor challenge [59,60,61,62]. He showed that such a challenge generates protective immunity shortly after tumor inoculation; he achieved this by demonstrating that the harvested lymphocytes of the tumor-bearing mouse could protect a naïve recipient against a lethal challenge. His further studies made a case that the protective cells were CD8 cytotoxic T cells [60]. In addition, he explored whether mice, given a lethal challenge, could come to harbor T cells that could suppress protective immunity. To this end, he gave normal and T cell depleted mice a normally lethal challenge. He showed that the administration of “protective cells”, at a substantial time after implantation, resulted in regression of the tumor implanted in T cell-depleted mice but not the tumor implanted in normal mice [60]. He further showed that the normal mice at this time harbored in their lymphoid organs T cells that could “suppress” (North’s word) the activity of protective cells [59,60,61,62]. His findings are summarized in Figure 2. North encapsulated his solution to the conundrum of why concomitant immunity does not protect against a lethal challenge in four words: “too little (concomitant immunity) too late”. It should be stated that North employed tumors of diverse type, and so it seems his findings might have broad significance. In addition, concomitant immunity was found to exist in many different tumor systems [58], a finding which also supports the potential generality of his findings.

## 10. Our Studies in North’s Systems Leading to the Th2-Skewing Hypothesis of Tumor Escape

I was intrigued by North’s conclusions for four reasons. Firstly, his depiction of concomitant immunity and its decay paralleled what Salvin and others had found regarding the expression of DTH when mice are given an intermediate dose of antigen; see Figure 1. Secondly, it seemed highly plausible that the generation of “North’s” suppressor T cells was associated with antibody production. This was made even more plausible when North identified the “suppressor cells” as CD4 T cells [60]. We had previously shown that the T cells associated with the production of antibody and able to inhibit the generation of DTH were CD4 T cells [5].

Thirdly, this possibility provided a natural explanation for the generalizations that were made in the 1960s and encapsulated by George Klein’s quote, which is cited above. This quote reflects the conclusion that cell-mediated immunity, not antibody, is protective against tumors [56]. Fourthly, it seemed this interpretation provided a natural explanation for the mechanism underlying “excision priming”. The tumor was excised at a time when concomitant immunity was optimally expressed, and when cell-mediated immunity had been optimally generated. The (partial) removal of the tumor at this time prevented the antigen-dependent evolution of the immune response towards a Th2, antibody mode [57]. The process of excision-priming thus seemed similar to the process Parish employed to establish a state of cell-mediated immune deviation [43,44]. We thought these observations could all be accommodated by the idea that the tumors grew progressively because they stimulated a response with a substantial Th2 component and the downregulation of the protective, cell-mediated response [63]. We wanted to test this idea. 

Given that tumor lines can change their properties, we were keen to use the lines North had himself employed. We were indeed fortunate that the Trudeau Institute, where North had worked, kindly provided these lines to us. Our studies, carried out primarily by Duane Hamilton, a PhD student, confirmed our conjectures and led us to propose the Th2-skewing hypothesis of tumor escape. Our principal findings [63] can be summarized. Tumor resistance and progression are, respectively, associated with stable Th1 responses and responses that develop a substantial or predominant Th2 component. Establishing a Th1 imprint, by a means modeled on Parish’s protocol of establishing cell-mediated immune deviation, resulted in resistance to a normally lethal challenge. An interesting technical aspect arose from these studies. We had to sacrifice mice when we wished to assess the Th1/Th2 phenotype of the anti-tumor response, employing an ELISPOT assay to enumerate antigen-specific, cytokine-producing cells [63]. We were very interested in longitudinal studies. We therefore explored whether we could indirectly follow the Th1/Th2 phenotype of the response by examining the IgG_1_/IgG_2_ ratio of the tumor-specific antibody. Although these assays were initially difficult to establish, we found them most useful once in hand. If we deliberately gave mice a number of tumor cells that resulted in a variable outcome, causing progression in some recipients but not in others, we could predict the outcome early after implantation on the basis of the IgG_1_/IgG_2_ ratio [63]. 

## 11. The Experimental Definition of N_t_, the Transition Number, and the Prospect of Universally Efficacious Vaccination against Entities Preferentially Susceptible to Cell Mediated Attack

Our hypothesis for the immunological correlates of tumor progression, for those tumors employed by North, make these situations, immunologically speaking, parallel to the situation in BALB/c infected with a number of *L. major* parasites resulting in progressive disease. We were inspired, in earlier studies, by Parish’s demonstration of cell-mediated immune deviation. We explored whether one could establish such a state in BALB/c in their response to leishmania parasites [52]. We found that infection of BALB/c mice with around 300 parasites resulted in a stable Th1 response and, in time, a Th1 imprint. Challenging these mice with a million parasites, the standard challenge that results in a Th2 response and progressive disease in naïve BALB/c mice, two months after the first infection resulted in a stable Th1 response and so in resistance. Such “Th1-imprinting” takes time to develop; the challenge must be administered a considerable time after the first infection to establish resistance, just as tumor resistance is only established some considerable time after tumor excision. 

A major question in the context of Salvin’s observations is their generality, a question also bearing on the generality and so usefulness of low-zone cell-mediated deviation. We explored the generality of Salvin’s observations in the mouse model of cutaneous leishmaniasis [52]. Briefly, we injected different strains of mice with different numbers of parasites, always by the same route, and followed the nature of the ensuing immune responses. We found in all strains of mice that infection with low numbers of parasites results in resistance and stable Th1 responses and with higher numbers, in time, in predominant Th2 responses and in progressive disease. We could define a transition number of parasites, N_t_, for each mouse strain. Infection with a number of parasites below N_t_ resulted in a stable Th1 response and with a number above N_t_ in a response that in time developed a Th2 component. Infection with a number considerably above N_t_ rapidly led to a predominant Th2 response. The value of N_t_ varied over a 100,000-fold range in the strains of mice we studied [53]. This finding leads us to suggest that Salvin’s observation represents a valid generalization, as it was true in all the strains of mice we employed. We also suggest the range in the value of N_t_ reflects genetic differences between the mouse strains [53]. In addition, it occurred to us that if a novel infection arises, such as HIV-1, and if there is usually a very great range in the value of N_t_ in responses to the newly appearing pathogen among the population, such diversity will result in diverse types of response and so may help us survive as a species. For example, most individuals infected by HIV-1 would, in the absence of treatment, seroconvert and suffer progressive disease. Only about 1% of those naturally infected, the elite controllers, generate a stable, cell-mediated response, which is associated with resistance [64]. It is known that only a few viral particles usually establish an infection [65], and so the value of N_t_ for HIV-1 infections must, in most people, be low [66]. The elite controllers will have relatively high N_t_s for this infection. If all individuals had low N_t_s, there would be no elite controllers. 

I believe there is a plausible and important inference arising from the very great variation in N_t_. The initiation of a response requires a minimal level of antigen-mediated CD4 T cell interactions, and the transition in the generation of Th1 to Th2 cells on a more substantial level of antigen-mediated CD4 T cell interactions. Most of the genetic variables that determine the dose of antigen able to support this substantial CD4 T cell interaction will also affect the level of antigen at which a minimal level of CD4 T cell interactions occurs. The great variability in the value of N_t_ implies, within the context of the concepts I discuss and favor here, that there will also be a great variability in the level of antigen that can initiate an immune response, i.e., a level that is immunogenic. This will be an essential consideration in some of the ideas developed below.

One reviewer thought it would be interesting if I speculated on what genetic loci might contribute to the variability of N_t_. Three, among others, seem particularly likely to be important to me. First, any host genes that affect the rate of replication of the invader. For example, different forms of the natural resistance-associated macrophage protein (Nramp) are known to greatly affect the growth rate of some intracellular parasites that replicate inside macrophages, such as *Leishmania donovani* and mycobacteria in the form of BCG. Such loci would affect the “antigenic level” of the invader. Secondly, it has always seemed to me that the predilection of mice of the C57Bl strain for generating cell-mediated rather than antibody responses is unlikely to be accidental, but may be related to the fact that these mice do not express functional IE class II MHC molecules. This presumably means they have considerably fewer CD4 T cells than if they had functional IE molecules. The threshold mechanism explains why such a relative deficiency would favor cell-mediated responses in place of antibody responses. Also, class I MHC molecules are also likely important, as cell-mediated immunity may be favored if the invader’s proteins are particularly well presented by the host’s class I MHC molecules. Lastly, loci that affect the nature of the T cell repertoire would likely be significant. A deficiency in this repertoire would likely favor cell-mediated over antibody responses. 

We have described elsewhere how the findings in the mouse model of cutaneous leishmaniasis may provide a basis for a strategy of universally efficacious vaccination against a pathogen uniquely susceptible to a cell-mediated attack [5,53]. The proposed strategy requires an attenuated, replicating organism that cross-reacts with the pathogen. It is therefore convenient to use BCG and vaccination against tuberculosis (TB) as the example. Immunization with a very low dose of BCG, below the transition number for any individual, is expected to result in an increased level of BCG in all individuals, in the absence of any immunity impeding BCG’s multiplication, until Th1 immunity is induced. In this way, a Th1 imprint can be generated in all inoculated individuals. I indicate below why I expect such a universal imprint to provide universally efficacious vaccination against TB.

## 12. The Implications of the Threshold Mechanism for Health

There are two main reasons for collecting the material in this essay in the way I have chosen. Firstly, the dose of antigen appears critical in determining the class of immunity generated. This dependence is puzzling in the context of a DAMP/PAMP-centric view, and so, because this view is so dominant, this fact appears to be largely ignored. I therefore wished to make the case for the importance of this dependence upfront. Secondly, I have found the concept of transition number, and an awareness that it can vary over a very wide range, has been fruitful in my thinking of diverse, medically significant situations. I would like to illustrate this significance in point form. It seems to me that these rather simple concepts are applicable in diverse situations. This gives me hope in the utility and in the validity of the potential insights.

(a)The basis of the treatment of human visceral leishmaniasis

We examined the IgG_1_/IgG_2_ ratio among leishmania-specific antibodies in three groups of seropositive individuals [67]. The first group were infected individuals that expressed strong DTH to leishmania antigens and were clinically diagnosed as healthy. These individuals are referred to as the healthy infected. The second group were patients recently diagnosed as ill and before the initiation of a standard 3-week treatment with a drug that kills the protozoan pathogen. We also examined the antibodies present in “cured” individuals after treatment. The IgG_1_/IgG_2_ ratio of patients was higher than that of the healthy-infected or the drug-cured individuals. The ratios in the latter two groups were indistinguishable. We suggested that the decrease in antigen load, as a consequence of treatment, modulated the insufficiently protective response, containing a significant Th2 component, into a predominant Th1 protective mode. It is known that treated patients are no longer infected in a manner that leads to disease. Their immunity is now protective. I should point out this is probably the only known and reliable treatment of a disease caused by a pathogen preferentially susceptible to cell-mediated attack. I think these findings would be much more widely appreciated if this disease were not uniquely tropical! It is natural to think such a process of treatment would be applicable to a pathogen present in the west and also preferentially susceptible to cell-mediated attack. I discuss this idea in the context of HIV-1 below. 

(b)Treatment of cutaneous leishmaniasis in the mouse model of the human disease

It is well known, as discussed above, that partial depletion of CD4 T cells, around the time of infection of BALB/c mice with a million parasites, results in a stable Th1 response, instead of the usual Th2 response, and so in resistance. Attempts to cure established infections by depletion of CD4 T cells were initially ineffective [50,51]. We surmised that this might be because, following the standard infection of BALB/c mice with a million parasites, a highly polarized Th2 response rapidly develops, which is not typical of most human disease. We infected BALB/c mice with 3000 parasites which resulted, in most mice, about two months post-infection, in a stable, mixed Th1/Th2 response and in a large and mostly stable lesion. We refer to such mice as having borderline disease. Partial depletion of CD4 T cells in these mice results in dramatic remission of lesions in a couple of weeks, and a modulation of the response towards a Th1 mode [5,49]. We suggest, partly on these grounds and on our observations described in the previous section on human visceral leishmaniasis, that on-going mixed Th1/Th2 responses can be modulated towards a Th1 mode by lowering the antigen level and/or reducing the number of CD4 T cells. We suggest on the basis of these observations that these variables of immunization that control the Th1/Th2 phenotype of primary responses also control the Th1/Th2 phenotype of on-going immune responses. We therefore employ these ideas as a working hypothesis in trying to understand how various treatments effect on-gong immune responses, as elaborated below. 

(c)Personalized immunotherapy of HIV-1 infections

We assume, for observational reasons already outlined, that predominant Th1 CTL responses are protective and can control HIV-1 infections. This idea incorporates the concept that this response can naturally increase in strength in time to catch up with and overtake the consequences of viral replication. This proposition is supported by the finding that all individuals who develop disease have seroconverted. There are no known long-term patients with only a cell-mediated response. 

Some well-established generalizations, as well as exceptions to them, have influenced our understanding of how immunity against HIV-1 is regulated. First, consider infected individuals receiving anti-retroviral therapy (ART) whose therapy is temporarily interrupted. It is most generally found that this results in a fairly rapid viral rebound [68]. This finding is usually taken to mean that ART is so successful at reducing the viral load that all immunity is lost. This seems plausible to me. However, in rare cases, there is no viral rebound and patients appear to have the immunity required to contain the infection [68]. What might be the basis of this resistance was unclear from the literature. However, we suggested that there is a plausible hypothesis within the framework developed above. 

Most individuals infected with HIV-1 and put on ART are seropositive and the anti-HIV-1 immune response probably already has a significant Th2 component. ART will reduce the viral load, leading to a modulation of the response towards an exclusive, protective Th1 mode. What is to be expected to result from further ART treatment? The viral load will presumably be further reduced until it reaches too low a level to sustain a cell-mediated response. In these cases, interruption of therapy results in viral rebound. This is the conventional explanation for viral rebound [68]. However, if interruption of therapy happens to occur at a time when a predominant Th1 response exists, viral rebound will not occur [66]. How can we judge when this propitious moment arrives?

We have discussed above how the class/subclass of antibody reflects the Th phenotype of the response. Some of the most dramatic evidence for such use of the IgG_1_/IgG_2_ ratio comes from HIV-1 infected individuals. Thus, even shortly after infection, and before and as seroconversion occurs, infected individuals have relatively mild symptoms, so this time is referred to as the honeymoon period. Symptoms of AIDS develop in most individuals after seroconversion, if untreated. Occasionally, seropositive individuals are also stable non-progressors [69,70]. Most interestingly, these individuals stably produce predominantly IgG_2_ anti-HIV-1 antibody, which is indicative of a stable and predominant Th1 response, as these researchers argue [69,70]. We have suggested that personalized immunotherapy is possible [66]. The IgG_1_/IgG_2_ ratio among anti-HIV-1 specific antibodies should be followed longitudinally as ART is administered. ART should be stopped once this ratio has become very small. We anticipate such individuals will control the levels of HIV-1, viral rebound will not occur, and these individuals will be relatively non-infectious [66]. 

To be honest, I had been thinking of this idea for several years. I was stimulated to write about it from further findings that were generally regarded as enigmatic but highly intriguing. They fit into the conceptual scheme we had developed. 

One study involved a group of HIV-1 infected individuals that had been on ART for three years, and for whom therapy was interrupted at this time. About 15% of the cohort had no viral rebound and were designated as “post-treatment controllers”. Thus, we have for natural HIV-1 infection about 1% of individuals who are elite controllers, and about 15% of individuals who are post-treatment controllers in a given situation [71,72]. Within the framework we have been developing, the elite controllers must have an exceptionally high N_t_, above the typical infectious dose of virus, for the reasons discussed above. We anticipate that the viral levels after three years of ART will be very low indeed. The virus is not permanently cleared because of its existence in a latent state [73] and its potential to be reactivated. Those individuals with the lowest N_t_ are expected, also for reasons outlined above, to be those able to still respond to the lowest levels of virus. These individuals are presumably the post-treatment controllers. We suggest our personalized treatment, outlined above, will allow the other 84% of HIV-1 infected individuals to gain control of their HIV-1 infections [66]. 

(d)Immune correlates of protection and disease upon infection by *Mycobacterium tuberculosis*: implications for the basis of failure to control the pathogen 

We examined the IgG_1_/IgG_2_ ratio of anti-mycobacterium-specific antibodies among a substantial number of TB patients and healthy healthcare workers who looked after TB patients. All these healthcare workers were positive in the PPD skin test and were therefore assumed to be infected, though healthy [74]. This ratio varied among healthy healthcare workers, whom we regarded as the healthy infected, from 0.001 to 1, and among TB patients from 0.001 to 100. It seemed obvious to us that those patients, with IgG_1_/IgG_2_ ratios above those found in the healthcare workers, had a greater Th2 component to their response than that existing in the healthy infected. It seemed they were somewhat like patients with visceral leishmaniasis; they presumably had a response with a substantial Th2 component that downregulated a protective Th1 response. We refer to these individuals as having type 2 tuberculosis The question arose as to why some individuals infected by the pathogen and who exhibited a response identical in type to that of the healthy infected, a predominant Th1 response, were ill. We refer to such patients as having type 1 tuberculosis.

Consider those individuals that develop type 2 tuberculosis. Given that the typical infectious dose not probably vary that much [75], these individuals will tend to have a relatively low value of N_t_ for their response against the pathogen, thus generating a response that develops, with time, a substantial Th2 component on exposure. Consider two individuals, one with an N_t_ for the pathogen that is moderate, M, and the other with a high N_t_, H. Given the range in values of N_t_ are likely considerable, M and H may differ by at least 100-fold. We have argued above that the limiting dose of antigen immunogenic for a Th1 response is most likely related to N_t_. Those with lower N_t_s will be able to respond to lower levels of the pathogen. Consider what a successful immune response must accomplish. It must kill the pathogen at some point, the “turning point”, at a slightly greater rate than the pathogen is increasing in numbers through replication. The individual with an N_t_ of M will start generating a Th1 immune response when the pathogen load is moderate, whereas the individual with an N_t_ value of H will likely only start generating such a response when the pathogen load is high. It is natural to suppose that the individual with an N_t_ of M will most often contain the infection in time, whereas the response of the individual with an N_t_ of H will take much longer after infection, or may *never* generate a response able to contain the much larger pathogen burden from which this individual suffers. This individual would have type 1 tuberculosis. 

I think it might be helpful if I try to encapsulate the above considerations. We assume the value of N_t_ has a wide range among individuals for infections by *Mycobacterium tuberculosis*. Over time, those with the lowest N_t_s, below a typical exposure dose, generate an immune response with a substantial Th2 component, associated with the downregulation of a protective Th1 response, and consequently develop type 2 TB. Individuals with a moderate N_t_ start to generate an immune response shortly after infection. This response contains the pathogen at an early time point. Such individuals constitute the healthy infected. Those individuals with a very high N_t_ do not start generating a protective Th1 response until the pathogen load has become very large, and only exhibit a response that can contain the pathogen after a prolonged period, if ever. Such individuals suffer from type 1 TB.

I found the TB literature uniquely frustrating over the years, as clear statements could not be made and so many enigmas existed [76]. Only through such experiences can one truly appreciate the value of an integrated framework. I conclude this section with a description of some of the enigmas in the TB literature and how I think our framework resolves them. I do so because thinking about the dynamics of pathogen growth and of immune responses against the pathogen have considerably influenced the way I think about immunity against cancer. 

Perhaps the biggest impediment to progress in the TB field was the inability to define immunological parameters that distinguished between the immunity of patients and of the healthy infected. This inability contrasts with the ability to define such parameters in the leishmaniases, for example. This inability can be understood in the context that disease can develop as a consequence of one of two types of failure by the immune system: the failure to generate a sufficiently large Th1 response to contain the pathogen, resulting in type 1 TB, due to the low immunogenicity of the pathogen in the infected individual; and failure to generate a sufficient Th1 response because of a substantial Th2 component that downregulates the Th1 component [74].

One of the paradoxes of lung TB, the most common form of the disease, was the apparent role of Th1 responses in lung granuloma formation, leading to pathology, and their presumed role in protection. Could the same response be protective and pathological at the same time [76]? Can protection be increased without increasing pathology to unbearable limits? We address this question shortly. Lastly, an important clue to curing the disease seemed to be the spontaneous cure of patients in sanatoria before the age of antibiotics [77]. These individuals, who undergo spontaneous cure, have all the hallmarks anticipated of a patient with type 1 TB. This course of events can be explained if the immune response manages to eventually reach the turning point but takes a considerable time, after infection, to do so. Moreover, the nature of these spontaneous cures suggests irreversible damage has not occurred to the lung during granuloma formation even in severely ill patients. This should give one confidence that increasing the Th1 response in patients with type 1 TB will not cause irreversible damage; a complete cure seems feasible. 

(e)Treatment of TB

In the late 1800s, Robert Koch prepared a protein extract from *M. tuberculosis* bacilli and administered it to TB patients in the hope of boosting the patients’ immunity and so realizing a resolution of the disease [78,79]. His “treatment” seemed to improve the status of some patients and make other patients more ill, sometimes even resulting in death [80]. Koch had to abandon this experimental approach to treatment. However, his findings can be naturally accounted for by our hypothesis of there being two types of tuberculosis [81]. We anticipate the Th1 response is insufficient to contain the pathogen in type 1 patients, and the level of antigen is insufficient to optimally stimulate the Th1 response. The administration of benign antigen would stimulate an increase in the size of the response without increasing the task the immune response would face if the pathogen load was increased. However, the administration of antigen to patients with type 2 TB would further the polarization towards the Th2 mode, and so further reduce the protective Th1 response. This is anticipated to be detrimental [81].

We have suggested, on this basis, that immunotherapy of TB may be realizable by optimally harnessing the patient’s own protective immunity. Patients would first be assessed to determine whether they have type 1 or type 2 TB, based primarily on the value of their IgG_1_/gG_2_ ratio. Type 2 patients would be treated with antibiotics and their IgG_1_/IgG_2_ ratio monitored longitudinally. When it was very low, antibiotic treatment would be stopped and clinical signs of improvement would be longitudinally assessed. We anticipate this would constitute effective treatment. Type 1 patients would be treated essentially as envisaged by Koch, and their clinical symptoms/parameters would be monitored, including their IgG_1_/IgG_2_ ratio. Clearly, it is important to ensure that treatment involving the administration of antigen does not result in polarization of the response towards a Th2 mode [81].

It is unclear for how long such treatments would need to be administered to be efficacious. The standard treatment of visceral leishmaniasis is a few weeks. I see no reason why a similarly short time would not be effective in treating most patients with type 2 TB. Clinical studies would help us understand how quickly treatment of type 1 TB might be efficacious. It is possible effective therapy would take only a matter of weeks. The implementation of this strategy would change the worldwide management of TB. Moreover, effective treatment is expected to reduce transmission and so break the epidemic cycle. Antibiotic therapy as currently practiced is ineffective unless sustained for about a year. This is most likely because it reduces the bacterial load to such an extent patients that any protective immunity is weakened or lost, and so treatment, to be effective, must result in the death of every last pathogenic organism. Combined antibiotic and antigen administration is likely to provide the most effective therapy of type 1 TB. I have always stressed a fact to my graduate students, as it provides such grounds for optimism. Although TB is one of the worst scourges of all infectious diseases, about 95% of people infected with the pathogen do not fall ill, as they naturally generate a protective response! We only need to tip the balance.

## 13. Prevention and Treatment of Human Cancer

(a)Prologue

My choice to consider human cancer at the end of this article does not reflect either my assessment of its general importance or the significance I place on the considerations elaborated upon here. It can be anticipated, given the wide genetic diversity of people, as well as the diversity of cancers, that it is difficult to make valid generalizations about immune responses against human cancers from observations made in this field alone. If the possibility is granted that some generalizations concerning how immune responses are regulated may not be unique to the field of cancer immunology, but may reflect ones that also hold in related fields, then it is natural to consider whether generalizations made in such fields, in which the diversity of the invader and/or the diversity of the host can be avoided, may be useful, hence my initial focus in writing this. This consideration has also naturally been a focus in my thinking.

(b)Current treatment of human cancer

I also would like to complement the narrative, given above, on the *history* of tumor immunology, as studied in animal models of human cancer, with some observations on current treatment of human cancer and research pertinent to improving such treatments. 

There seems to be an agreement from both human [82] and animal studies [60,61] that cytotoxic T cells are the most effective cells for containing cancers and tumors. Given the restrictions on human studies, the most compelling evidence comes from correlations between the prevalence of CD8 T cells and clinical outcome. Accepting this as a plausible, working assumption, we can imagine at least two “immunological reasons” why a cancer is not contained. Firstly, the cancer may be insufficiently immunogenic, so the CTL response generated is too weak to kill the cancer cells faster than they increase in number by division. Secondly, the CTL response may be too weak because the immune response has deviated into a mode other than that which is optimally protective, and so the protective response is downregulated. Establishing a Th1 imprint specific for antigens commonly associated with a type of cancer should provide protection against this type of cancer. The rationale in this case is the same as that which explains why a Th1 imprint would protect against both type 1 and type 2 TB [81].

Most treatments are designed to get rid of cancer cells: surgery, the use either of radiation and/or chemotherapy that kills cells bent on dividing, and immunotherapy, which we shall discuss shortly. Radiation and chemotherapy obviously may also affect the cells of the immune system that divide, and surgery, radiation and chemotherapy also reduce antigen load and, in view of the considerations outlined above, may well affect both the class of immunity generated and its strength.

(c)Modern developments pertinent to cancer immunotherapy

Four kinds of observations/developments over the last few decades, concerning human cancer, seem particularly significant to me. Several researchers, particularly Thierry Boon and his colleagues, have cloned cancer-specific T cells from cancer patients and used these T cell clones to define antigens recognized by the patient’s T cells [83,84]. Such studies have shown that there are often common antigens recognized by different patients’ T cells with a common type of cancer. Secondly, many patients produce antibodies that can react with the cancer cells [85]. Thirdly, one must note the remarkable investigations employing T cells artificially endowed with chimeric TcR that can recognize antigens present on tumor cells [86]. These striking studies are motivated by the desire to generate in a patient effective immunity de novo, in contrast to understanding how the patient’s natural, protective response is regulated, and so how it might be harnessed. I share with North what I imagine was his hope: understanding what constitutes effective immunity, and how its expression is naturally regulated, will give rise, in the long run, to the most efficacious and gentlest forms of treatment. I do not discuss chimeric TcR therapy further, given this focus. Fourthly and lastly, once T cells are activated, their further activation is inhibited by immune checkpoint inhibitors; these are mediated by receptors on T cells that block costimulatory signals and may also mediate negative signals on interacting with their ligands. Most prominent among these checkpoint receptors on T cells are CTLA4 and PD1. Many studies have shown that, in some cancer patients, blocking checkpoint inhibitory signals can increase patient survival and even result in recovery [87,88]. How these function, when effective, is presumed to be by removing signals that inhibit further T cell activation, and so increasing the intensity of T cell responses. Why such therapy is sometimes effective and sometimes not is unclear. In addition, some of these treatments have significant side-effects [87,88]. 

(d)Centrality of monitoring the nature of the anti-cancer immune response

It seems essential, if we are to make immunotherapy more rational and effective, to have readily available assays to assess the nature of the immunity against the cancer in particular patients, should this be possible. Given the definition of antigens commonly present in cancers of similar type in different patients [83,84], it appears it would not be too difficult to develop standard ELISA assays, employing a collection of these antigen for a particular type of cancer, to measure in a standard fashion the relative presence of different classes/subclasses of antibodies to these antigens. The class/subclass of antibody to the different antigens of the cancer are expected to be the same/similar, due to the “coherence” of the class/subclass of antibody generated against different antigens belonging to the same cell [89]. Thus, the antigens employed in the ELISA should contain some but need not contain all the cancer antigens responded against. As such ELISA assays seem feasible, I assume in the considerations below their eventual availability. 

(e)Improving current treatment

We shall only gain really plausible insights into how immune responses against human cancer are regulated once we have found reliable means to manipulate these responses to achieve effective immunotherapy. The following ideas are put forward in this spirit.

Various observations support the possibility that the Th2-skewing hypothesis is pertinent to some human cancers [90,91,92,93,94]. Given that surgery, irradiation, and chemotherapy are expected to reduce the antigen load, and that such a reduction is expected to affect the Th1/Th2 phenotype of the anti-cancer immune response, it would seem that longitudinal monitoring of the Th1/Th2 phenotype of the response, via the IgG_1_/IgG_2_ ratio methodology, during such treatment, might be invaluable. For example, if this ratio was high at initiation of treatment of a significant fraction of cancers, it would indicate the pertinence of the Th2-skewing hypothesis; if, in these cases, the ratio became lower as treatment progressed, it would provide evidence that treatment affects the Th1/Th2 phenotype of the immune response. In such circumstances, it might be ethical to interrupt a course of treatment if the IgG_1_/IgG_2_ ratio became sufficiently small in a manner known to reflect a predominant Th1 response. Further chemotherapy at this stage might undermine the protective response. 

The effects of radiation and chemotherapy would likely affect the Th1/Th2 response by other means than just killing cancer cells and so reducing the antigen load. They result in the killing of dividing cells, other than cancer cells. The administration of cyclophosphamide, a chemotherapeutic agent, for example, results in the alkylation of DNA of cells and causes their death if they are forced into cell division. Cyclophosphamide, when administered close to the time of immunizing with non-replicating antigens, is known to cause a modulation of an immune response from a humoral to a cell-mediated phenotype [95]. North showed that administration of a similar dose of cyclophosphamide before administration of a normally lethal challenge of a tumor resulted in resistance [96], as did the (partial) depletion of CD4 T cells [97]. All these observations find a natural explanation in terms of the threshold mechanism. 

Examining the Th1/Th2 phenotype of the anti-cancer response before initiating a therapy involving blockers of checkpoint inhibitors, might also be enlightening. It would make sense if this treatment was more effective in cases where the Th1 protective response was weak, due to the weak immunogenicity of the cancer, rather than in a situation where the response was ineffective due to having too significant a Th2 or other Th component, associated with downregulation of a protective Th1 response. 

The administration of blockers of checkpoint inhibitors to cancer patients can, in some cases, have undesirable side effects [86,87]. It is perhaps inevitable that the administration of molecules that are not designed to affect only specific immune responses will have side effects. If it is really the case that responsiveness to checkpoint therapy is associated with intrinsically weak Th1 and CTL responses, it may be possible to increase the intensity of such responses by administering antigen, as described above in the immunotherapy of type 1 TB. This approach would, presumably, have fewer side effects.

(f)Epilogue: The international panel on immunotherapy of HIV-1 infections

I do not know how immunologists and cancer immunologists will respond if they read this article. I am sometimes afraid they will tire of the description of all those old studies and find the lack of detailed molecular considerations a sign of unprofessionalism. I could not help becoming more interested in the sociological aspects of performing science over the years as I became more experienced and puzzled by what appeared to me to be the lack of focus on basic questions and in seeking generalizations of broad validity. It is only natural to hope that one’s research will not only provide insight but will be helpful to society in solving practical problems. I have been interested in how AIDS might be prevented and treated for over three decades. I am sure many share such aspirations. My style in writing this article was driven by what I consider to be research silos in contemporary science. A meeting in 2015 was held to discuss how the HIV-1 epidemic might be addressed, and a report was made [98]. There was no mention made in this report that the antigen dose affects the type of immunity generated. Nor was there much consideration of whether immunity to other infections might provide clues as to how to curtail the HIV-1 epidemic. I become motivated, when I read such communal efforts, to write such articles as this one! 

## 14. Conclusions

A framework is proposed that accounts for how antigens interact differently with the cells of the immune system to result in the generation of Th1 or Th2 cells, and so in different classes of immunity. This Threshold Hypothesis was initially based on studies carried out with simple antigens, such as purified proteins, and directed at analyzing how such immune responses are regulated. The proposed threshold mechanism is quantitative. The framework this hypothesis provides can be used to understand the regulation of immune responses to more complex antigens in animal models of infectious disease and of cancer. This includes the successful development of strategies to prevent or treat disease caused by agents preferentially susceptible to cell mediated Th1 attack. We discuss whether and how strategies of prevention and treatment of human cancer and particular infectious diseases may be possible along similar lines. Although speculative, such proposals make clear and precise predictions and so their feasibility and value are readily testable. 

## Figures and Tables

**Figure 1 cancers-16-01431-f001:**
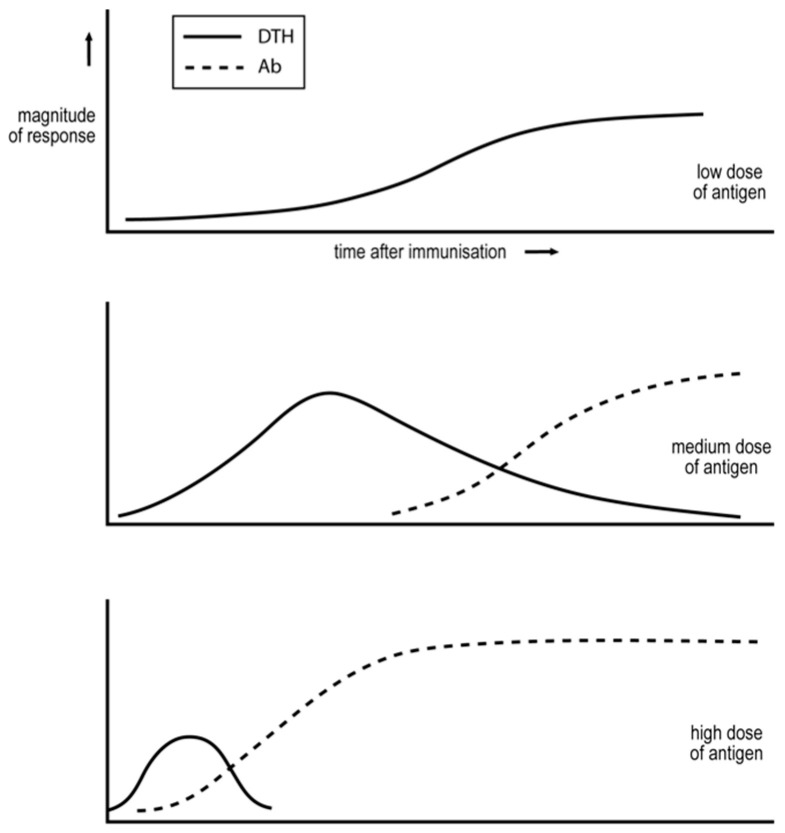
Variables of immunization that affect the DTH/IgG antibody nature of the immune response. The horizontal axis represents the time after antigen impact. Adapted from [41].

**Figure 2 cancers-16-01431-f002:**
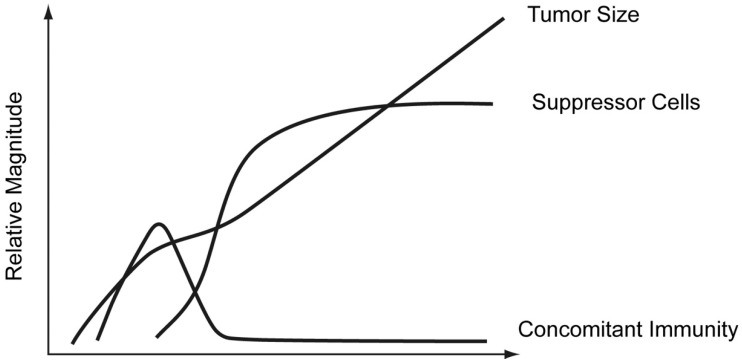
North’s picture of the kinetics of tumor growth, as well as expression of concomitant immunity, and of T cells able to suppress this immunity, in mice implanted with a lethal dose of tumor cells. Adapted from [41].

## Data Availability

All pertinent data can be found in the references. No unpublished data are referred to.

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
