# Peer review of "A Plausible Framework Reveals Potential Similarities in the Regulation of Immunity against Some Cancers and Some Infectious Agents: Implications for Prevention and Treatment"

_cancers, 2024, doi:10.3390/cancers16071431_

Round 1

Reviewer 1 Report

Comments and Suggestions for Authors

This manuscript is a brilliant piece of scientific writing that showcases a deep understanding of the complex mechanisms that regulate the immune system, highlighting their inconsistencies with classical observations. It introduces the innovative Threshold Hypothesis as an alternative framework that effectively addresses these paradoxical observations. By exploring its applicability beyond its original scope, particularly in the realms of cancer immunity and defense against infectious agents, the Threshold Hypothesis emerges as a powerful tool for elucidating common immune mechanisms and proposing novel strategies for preventing and treating cancer and infectious diseases. This holistic approach underscores the significance of fundamental immunological principles, showcasing the potential of the Threshold Hypothesis to revolutionize immunotherapy and disease management. The writing is engaging and easy to follow, with a logical flow of ideas. Overall, this manuscript is a testament to the author's expertise in the field and the ability to communicate complex scientific concepts in a clear and concise manner. I recommend this manuscript to be accepted for publication.

Author Response

I thank the reviewer for their encouraging comments. I am grateful that the reviewer was open to consider what I feel are paradoxes of the field. Thank you.

Reviewer 2 Report

Comments and Suggestions for Authors

In the current study, the author discussed a plausible framework about the regulation of immunity against some cancers and some      infectious agents. And the author also proposed non-invasive and efficacious strategies to prevent and treat cancer and infectious diseases caused by pathogens uniquely susceptible to cell-mediated immunity. Overall, this review study is very enlightening and instructive. The paper is well written as well. I believe, to some extent,  this paper will benefit the field in the future.

Author Response

I thank the reviewer for their encouraging comments. 

Reviewer 3 Report

Comments and Suggestions for Authors

The perspective by Peter A Bretscher proposes the Threshold Hypothesis and a comprehensive framework aimed at elucidating the regulation of immune responses, particularly the impact of antigen levels on the generation of Th1 or Th2 cells, thereby influencing cell-mediated immunity against infectious diseases and cancers. Through an exhaustive analysis and discourse on established generalizations and classical yet paradoxical observations regarding Th1 and Th2 responses upon antigenic stimulation, the author underscores the applicability of the proposed framework in understanding immune response mechanisms and its potential for enhancing cell-mediated immunity for improved treatment outcomes.

The perspective is impeccably written and highly engaging. I only have a few minor comments:

1.       In the summary and abstract sections of the perspective, a more direct description of the antigen level-dependent Th1/Th2 responses could enhance clarity and appeal to readers.

2.       The authors delve into the variation of Nt among different individuals and its impact on pathogen responses, leading to diverse Th1 or Th2 responses. I am wondering whether the author has any hypothesis about the factors causing the variations in Nt among individuals.

Author Response

I thank the reviewer for their sympathetic and careful reading of the manuscript. The reviewer made two suggestions. He/she suggested that I would make the Summary and Abstract more appealing if I indicated how antigen dose affects the cell-mediated/antibody nature of the ensuing response. I realised, on reflection that I had misunderstood the first suggestion made by this reviewer in my initial response. I apologize. I therefore  withdraw  my initial response to the first suggestion. In responding positively to the advice, I have added one sentence ("in red") to each of the Summary and the Abstract, indicating that low and higher doses of antigen respectively induce cell-mediated and IgG antibody responses.

     The reviewer also suggested it would be interesting to speculate on what genetic loci contribute to the great variability in Nt that we find. I have added a paragraph in response to this suggestion, around line 500, and given in red. Here it is:

One reviewer thought it would be interesting if I speculated on what genetic loci might contribute to the variability of Nt. Three, among others, seem particularly likely to be important to me. First, any host genes that affect the rate of replication of the invader. For example, different forms of the natural resistance associated macrophage protein (Nramp) are known to greatly affect the growth rate of some intracellular parasites, that replicate inside macrophages, such as Leishmania donovani and mycobacteria in the form of BCG. Such loci would affect the "antigenic level" of the invader.  Secondly, it has always seemed to me that the predilection of mice of the C57Bl strain for generating cell-mediated rather than antibody responses is unlikely to be accidental, but related to the fact that these mice do not express functional IE class II MHC molecules. This presumably means they have considerably fewer CD4 T cells than if they had functional IE molecules. The threshold mechanism explains why such a relative deficiency would favor cell-mediated in place of antibody responses. Also, class I MHC molecules are also likely important, as cell-mediated immunity may be favoured if the invader's proteins are particularly well presented by the host’s class I MHC molecules. Lastly, loci that affect the nature of the T cell repertoire would likely be significant. A deficiency in this repertoire would likely favour cell-mediated over antibody responses.